# Relationship between surface deformation displacement and energy evolution of red sandstone under uniaxial compression

Feng Gao[1,2], Guangjun Cui[1], Jin Liao[1,2], Zhen Liu[1,2]*, Cuiying Zhou[3]*, Chunhui Lan[1,2], Ziyu Tao[1,2]

1 Guangdong Engineering Research Centre for Major Infrastructure Safety, Sun Yat-sen University, Guangzhou, China, 2 School of Civil Engineering, Sun Yat-sen University, Guangzhou, China, 3 School of Civil and Transportation Engineering, Guangdong University of Technology, Waihuan West Road, Guangzhou University Town, Guangzhou, China

* liuzh8@mail.sysu.edu.cn (ZL); zhoucy@mail.sysu.edu.cn (CZ)

## Abstract

The bedding structure, variations in cementation state, and pore network characteristics of red sandstone contribute to significant non-uniform deformation, critically influencing engineering stability. Existing research on external energy input to red sandstone primarily focuses on the relationship between overall rock deformation, axial deformation, and energy evolution, while studies exploring the correlation between red sandstone surface deformation caused by loads change and energy remain limited. To address these gaps, this study employs a 3D visualization test system to collect surface displacement data of red sandstone standard samples synchronously using multiple cameras, combined with micron-scale digital image correlation technology and energy analysis. The study systematically examines the relationship between surface deformation and energy evolution in red sandstone and reveals the surface deformation process of red sandstone. According to the surface deformation's variation trend of the sample and the degree of change, four distinct stages of surface deformation, corresponding to axial deformation stages but with temporal discrepancies, are identified. By quantifying external energy input, the study establishes a correlation model linking obtained local surface deformation to energy input under unconfined conditions. The findings provide critical theoretical and technical support for stability evaluation and optimization design of red sandstone engineering, especially in deformation control and safety warning for slopes and other free-surface engineering projects.

## 1 Introduction

Red sandstone is widely distributed, with most of the engineering projects either directly or indirectly constructed in red sandstone regions. Consequently, in

**Data availability statement:** All relevant data are within the paper and its Supporting Information files.

**Funding:** Funding The research is supported by the National Natural Science Foundation of China (NSFC) (Grant No. 42293354, 42277131, 42293351, 42293355, 42293350). The funders had no role in study design, data collection and analysis, decision to publish, or preparation of the manuscript. There was no additional external funding received for this study.

**Competing interests:** The authors have declared that no competing interests exist.

excavation-based projects such as slopes, tunnels, etc., the properties of red sandstone such as deformation and failure significantly influence overall project stability and safety [1–4]. Current research on rock deformation predominantly analyzes and explains the process through crack propagation and energy evolution. However, most studies focus on either microscopic surface crack formation or overall rock deformation, failure, and energy evolution, with limited investigation into the relationship between surface deformation and energy input [5–7]. The energy input resulting from construction disturbance (such as tunneling and pile foundation construction) inevitably induces deformation and failure, potentially compromising project safety [8]. Developing a correlation model between energy input and surface deformation could enable more accurate predictions and evaluations of the deformation behavior of red sandstone, providing a theoretical basis for stability control. Given these considerations, there is an urgent need to analyze the relationship between red sandstone surface deformation and external load or energy input. Investigating red sandstone surface deformation from an energy evolution perspective allows for the establishment of a correlation between red sandstone surface deformation and external energy input. In engineering applications, such as tunnel excavation, slope excavation, pile foundation construction, etc., the deformation within the scope of construction disturbance can be analyzed and predicted to ensure project safety.

Deformation research primarily relies on experimental measurements, while energy evolution analysis integrates theoretical research with experimental validation. Studies on red sandstone deformation and energy evolution in slopes and can be categorized into three main approaches: experimental research, theoretical analysis, and numerical simulation. Experimental studies analyze the non-uniform deformation process of rock-like materials by measuring the evolution of the deformation field on the specimen surface during loading, and the parameters such as the width of the localized zone are quantitatively described [9,10]; the strain gauge method records red sandstone deformation and failure [11]; CT analysis [12]; Acoustic Emission method (AE) [13]; and Digital Speckle Correlation Measurement method (DSCM) [14] etc. The strain gauge method possesses the advantages of high sensitivity and dynamic response. The CT analysis method is capable of facilitating non-destructive testing and high-resolution imaging. The AE technology offers real-time monitoring and high sensitivity to damage. DSCM accurately quantifies rock deformation. In the realm of theoretical research, scholars have delved into the energy conversion laws governing rock failure by examining the energy evolution rate through the lens of elastic energy conversion rates, conversion functions, and multi-state constitutive models; they have also elucidated the energy evolution characteristics inherent in the damage process and revealed the correlation mechanism between the deformation characteristics of rock mass damage and energy consumption [6,15–17]. Concurrently, the mechanical essence of rock failure is recognized as an energy-driven instability process, transitioning from local damage to a comprehensive catastrophe, which is accompanied by a chain response of energy input, accumulation, and dissipation [18]. Scholars have systematically studied the energy evolution characteristics under varying loads, prefabricated cracks and strain rates, focused on the correlation

mechanism between rock deformation and energy dissipation, quantitatively revealed the evolution law of damage degradation and failure characteristics, and clarified the dynamic energy response mechanism underlying the instability failure process [19–22]. In numerical simulations pertaining to rock and soil, the Discrete Element Method (DEM) and the hybrid Finite Discrete Element Method (FDEM) (exemplified by software tools such as PFC2/ 3D, UDEC, ELFEN) are employed to characterize meso-scale defects, including rock grain boundaries and microcracks, through particle contact interactions. Conversely, continuum methods, such as the finite element method, utilize statistical distribution functions to represent the meso-mechanical properties of rock materials. These distinct approaches offer substantial advantages in simulating the progressive failure, instability, and large deformation of rock and soil [23–25]. Despite these advancements, existing red sandstone deformation monitoring techniques have limitations: strain gauges are susceptible to interference and cannot be reused; CT scanning has low efficiency; AE methods suffer from high environmental sensitivity; DSCM is restricted to surface displacement measurement; numerical simulation accuracy is affected by parameter uncertainties. Some scholars have studied the compression failure mode of defective rock mass under load through digital image technology, which is helpful to better understand the failure behavior of defective rock mass under compression shear load, this method also plays an important role in the deformation analysis of rock sample surface under energy input [2,26]. Furthermore, while extensive studies focus on the internal energy evolution of red sandstone, research on surface energy evolution remains insufficient. The relationship between local surface deformation and input energy remains unclear, and the understanding of red sandstone's surface deformation association under loading and other external energy input conditions remains limited.

Given these challenges, there is a pressing need for further investigation into the external energy input and surface deformation response model of red sandstone. Therefore, this study selects red sandstone as the specimen and employs the uniaxial compression test method. To observe the surface deformation of red sandstone, a high-spatial-temporal-resolution 3D visualization test system for multi-field coupling damage, integrated with energy analysis, is utilized. This research endeavors to elucidate the relationship between red sandstone deformation and external energy input, which holds significant implications for addressing red sandstone engineering problems, such as those pertaining to slopes.

## 2 Research content and methods

As a typical sedimentary rock, understanding the damage evolution of red sandstone is a key scientific foundation for ensuring the stability of surrounding rock in underground engineering. Most existing studies focus on the correlation between macroscopic axial strain energy and energy dissipation pathways in red sandstone, while research on the role of surface deformation in overall red sandstone stability remains insufficient. In particular, a systematic correlation between surface deformation and energy evolution has not yet been established. This study addresses these gaps by investigating sample preparation, experimental testing, and data analysis. Speckle processing is applied to the sample for image analysis preparation, followed by uniaxial compression tests to obtain stress-strain data. Simultaneously, surface deformation data are obtained using a high spatial and temporal resolution 3D visualization test system for multi-field coupling damage analysis. The collected data are processed and analyzed to determine the relationship between surface deformation and energy evolution in red sandstone. The specific experimental process is outlined in the following Fig 1.

### 2.1 Test contents

**2.1.1 Test procedure and scheme.** Experiments and data analysis are carried out using the high spatial and temporal resolution 3D visualization test system for multi-field coupling damage at Sun Yat-sen University [27]. This system provides data on stress-strain curve, dynamic Poisson's ratio, surface local deformation displacement and others.

The test system consists of several key components: Fully transparent water confining pressure chamber (Fig 3 (b)): Enables Digital Image Correlation (DIC) technology to capture crack initiation and propagation on the surface of red sandstone samples, allowing simultaneous testing of four groups of uniaxial, triaxial or rheological rock samples under different

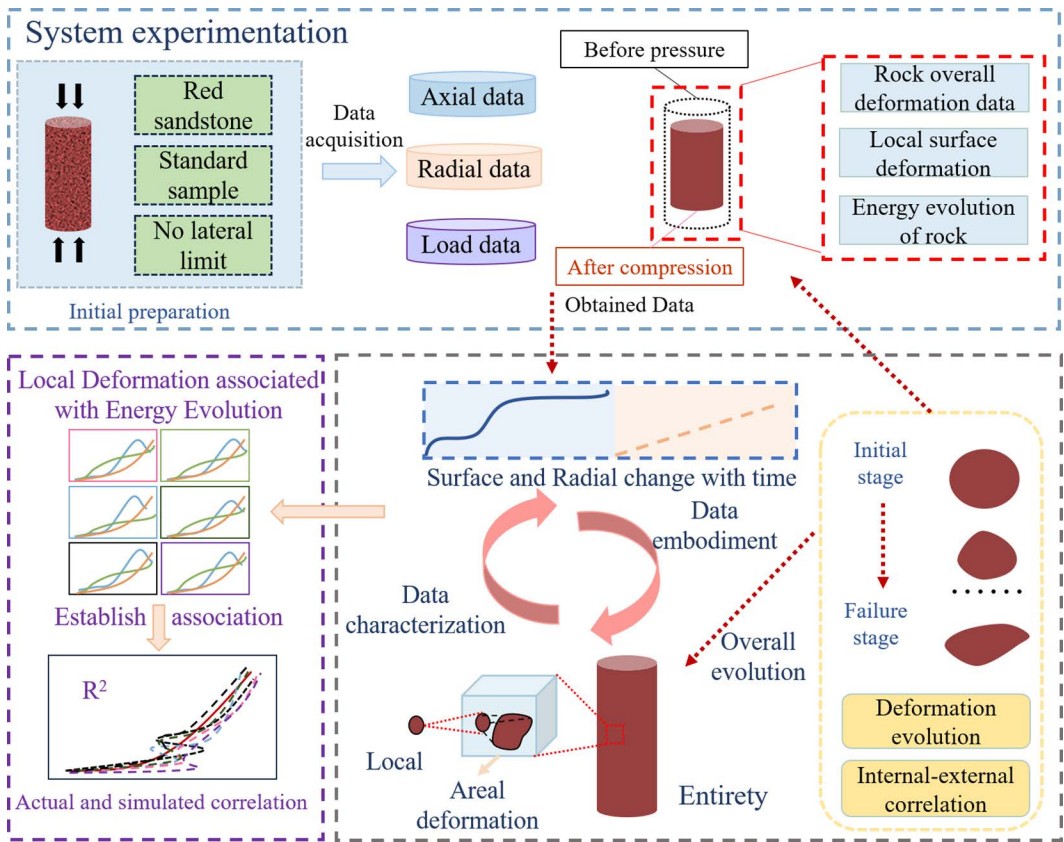

**Fig 1. Test flow chart.**

loading conditions; Damage high-resolution 3D imaging module (Fig 3 (b) (c)): Monitors surface damage of rock samples using DIC with a spatial resolution of 5μm and a temporal resolution of 1ms. A built-in digital speckle imaging quality quantitative evaluation system is employed to ascertain whether the above speckle quality meets the requisite accuracy standards; Rapid data synchronization and processing module: Facilitates synchronous processing of macro and micro data; Eight-camera system (Fig 2 (a) (b)): Consists of four pairs of adjacent cameras, providing 360° monitoring of rock specimen surfaces.

The eight-camera system reconstructs the full-field displacement field during red sandstone compression deformation, using point cloud synthesis method to fuse multi view data to obtain three-dimensional strain distribution of sample surface. Given the anisotropic deformation characteristics, four regions (C1–C4) are selected (Fig 4) to extract: Horizontal displacement (HD) (Fig 4c, x-axis); Out-of-plane displacement (OD) (Fig 4c, z-axis); Vertical displacement (VD) (Fig 4c, y-axis). These displacements characterize the multi-directional coupling deformation law [28].

**2.1.2 Test samples.** The mineral and oxide composition of red sandstone significantly influences its physical and mechanical properties, hardness, elastic modulus, strength and Poisson's ratio, which in turn affect its deformation and failure modes.

In Table 1, Qz CJ-1~CJ-6 accounted for 65.81~71.34%, combined with high $SiO_2$ content (78.18~87.87.41% in Table 2), it is consistent with the characteristics of sandstone dominated by quartz. All samples contain Hem, with contents of 6.41%, 11.41% and 3.25%, respectively. Hematite is a typical red iron oxide, and its presence directly explains the red appearance

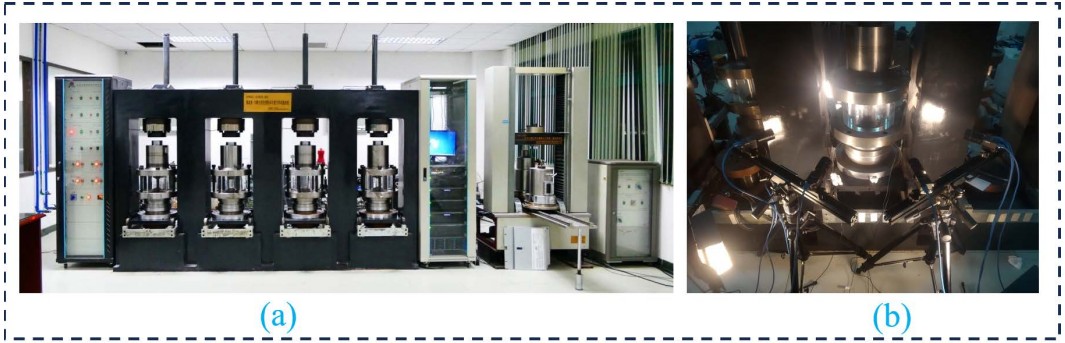

**Fig 2. High spatial and temporal resolution 3D visualization test system for multi-field coupling damage. (a)** System overview. **(b)** Test section.

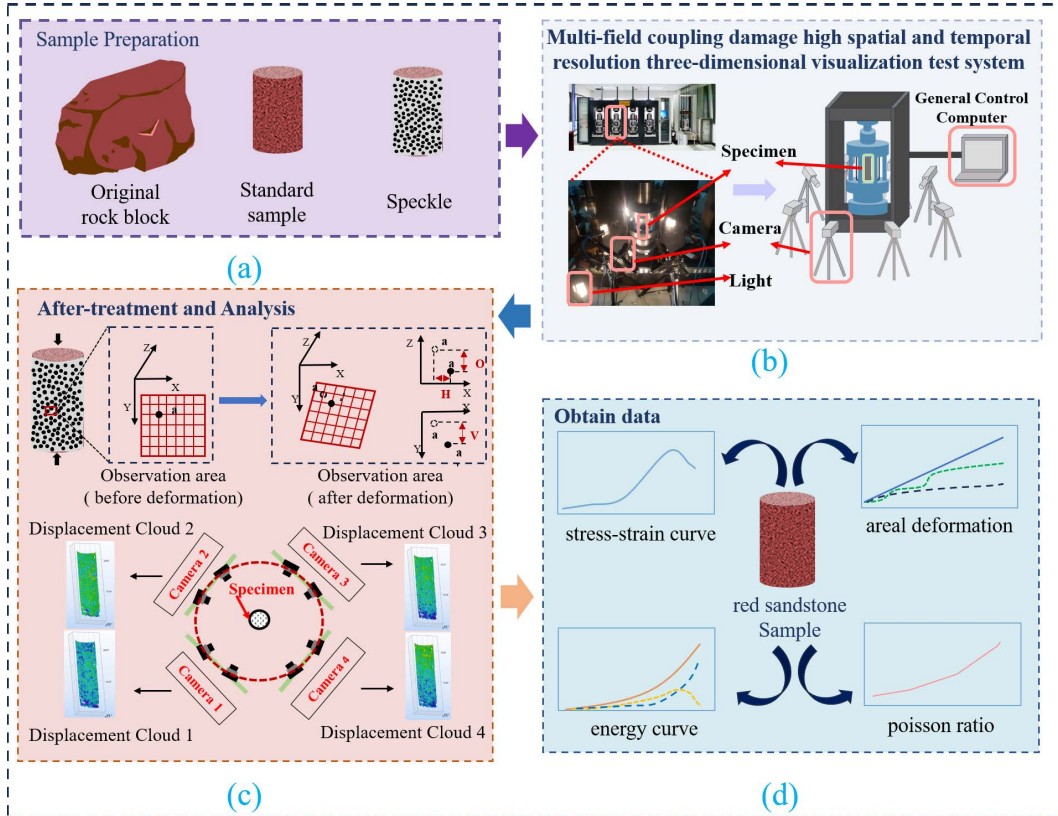

**Fig 3. Test system and treatment. (a)** Sample preparation. **(b)** Testing device. **(c)** Data processing. **(d)** Sample data.

of the rock. The content of $TFe_2O_3$ (total iron oxide) (0.74%~1.60%) in Table 2 further supports the existence of iron cement, which is consistent with the genesis of red sandstone.

According to ISRM standards, cylindrical samples are prepared with a height of 100 mm and a diameter of 50 mm for uniaxial compression test and image processing. Specimens were extracted from homogeneous blocks free of visible fractures. Cores were diamond-drilled perpendicular to bedding, machined to ISRM standards

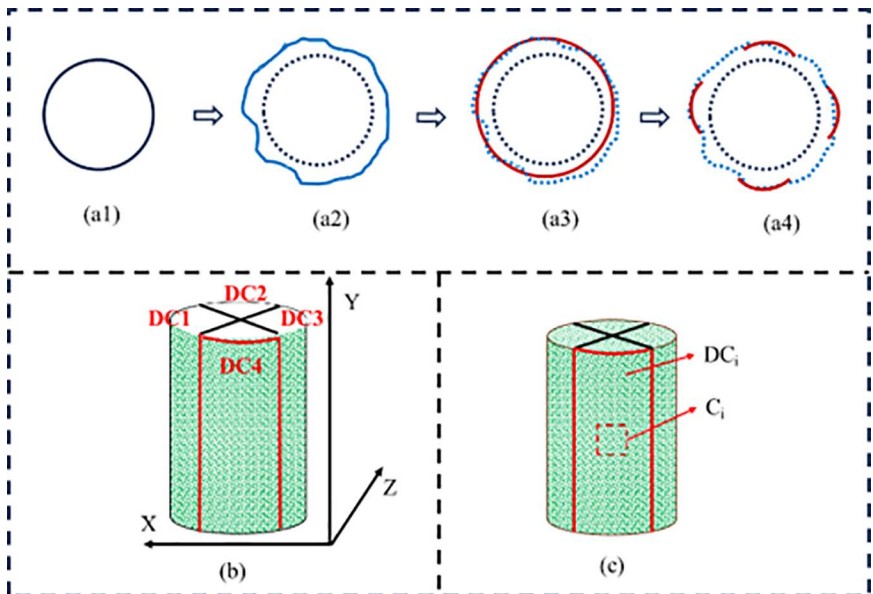

**Fig 4. Simplified model of sample surface deformation.**

**Table 1. Mineral composition of red sandstone.**

| Sample | Qz | Cal | Kfs | Ab | Ilt | Ms | Hem |
|---|---|---|---|---|---|---|---|
| CJ-1~CJ-2 | 71.34 | 1.17 | 5.76 | 15.33 | – | – | 6.41 |
| CJ-3~CJ-4 | 65.81 | 1.30 | 21.48 | – | – | – | 11.41 |
| CJ-5~CJ-6 | 28.03 | 1.21 | 20.64 | 14.41 | 7.78 | 24.69 | 3.25 |

**Table 2. Red sandstone oxide composition.**

| Sample | $SiO_2$ | $Al_2O_3$ | $TFe_2O_3$ | CaO | MgO | $K_2O$ |
|---|---|---|---|---|---|---|
| CJ-1~CJ-2 | 81.18 | 14.97 | 1.60 | – | 1.59 | 0.66 |
| CJ-3~CJ-4 | 87.41 | 7.42 | 0.74 | 1.47 | 0.96 | 2.00 |
| CJ-5~CJ-6 | 78.18 | 14.15 | 1.11 | 1.40 | 1.61 | 3.56 |

(flatness ≤ ±0.02 mm). Based on a review of digital image processing workflows from previous studies [29,30], it is imperative to initially apply speckle patterns onto the surface of red sandstone specimens. It is widely acknowledged that the quality of the speckle pattern serves as the foundation for the DIC method. High-quality speckle pattern is essential for high-precision measurements of displacement and strain. Some scholars have established pertinent standards for speckle patterns [31,32]. In this study, the speckle pattern is applied uniformly across the entire 360 degree surface of the red sandstone specimen, facilitating comprehensive monitoring of displacement and strain of the red sandstone surface during testing. The speckle pattern is printed on the specimen surface using high-precision, low-noise drum transfer printing techniques. The speckle size is set at 1 mm, which is optimal for 75 × 100 mm field of view, and the resolution is maintained at 5 million pixels to ensure superior speckle quality, as shown in Fig 5 below.

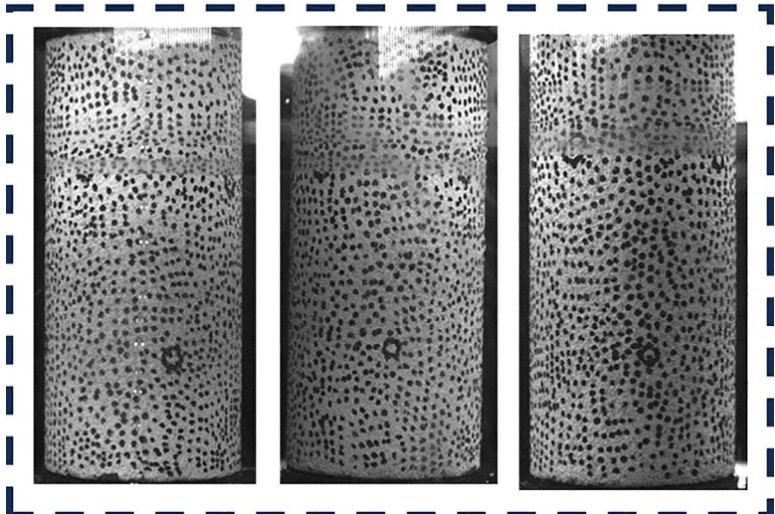

**Fig 5. Sample and speckle patterning.**

## 2.2 Research method of energy and surface local deformation of red sandstone

Red sandstone undergoes energy input, accumulation, dissipation, and release under external loading. By integrating the analysis of local surface deformation, it becomes possible to elucidate the correlation mechanism between energy and surface deformation of red sandstone under applied loads.

### 2.2.1 Red Sandstone energy analysis.

1. Total energy

The testing apparatus performs energy input by applying an axial load. Red sandstone progressively accumulates energy in the form of elastic strain energy and dynamically dissipates and releases energy during crack propagation. Notably, elastic strain energy exhibits reversibility under certain conditions, whereas dissipated energy follows a unidirectional and irreversible path under all conditions. The energy calculation process under uniaxial compression is outlined as follows. [6]:

$$U_0 = U_e + U_d \tag{1}$$

$$U_0 = \int_0^\varepsilon \sigma_1 d\varepsilon_1 \tag{2}$$

$$U_e = \frac{\sigma_1^2}{2E} \tag{3}$$

$$U_d = \int_0^{\varepsilon_1} \sigma_1 d\varepsilon_1 - \frac{\sigma_1^2}{2E} \tag{4}$$

Where $U_0$, $U_e$ and $U_d$ are the total absorbed energy, stored elastic strain energy, and dissipated released energy of red sandstone, respectively, $\sigma_1$ and $\varepsilon_1$ are axial stress and axial strain respectively.

## 2. Energy ratio and dissipation rate

The total energy of red sandstone is influenced by mineral composition and pore structure but does not directly indicate internal energy conversion efficiency. The energy ratio and dissipation rate reflect the internal efficiency of red sandstone, reflecting the processes of fracture expansion and frictional heat release. These metrics provide insights into the dynamic process of energy storage, distribution, and dissipation within red sandstone under applied loads. Specifically, the ratio of dissipated energy to elastic strain energy is defined as the quotient of dissipated energy to elastic strain energy $K_{d/e}$ at a given instance during the loading and failure process of the specimen. This ratio quantitatively describes the energy conversion occurring at each stage of failure, thereby revealing the energy flow patterns. The dissipation rate $\nu_{U^d}$ is employed to characterize the dynamic failure process during the loading process. This rate is determined by the magnitude of the derivative of the dissipated energy density with respect to time, reflecting damage severity. [33]:

$$K_{d/e} = \frac{U_d^t}{U_e^t} \tag{5}$$

$$\nu_{U^d} = \left| \lim_{\Delta t \to 0} \frac{\Delta U_d}{\Delta t} \right| = \left| \frac{dU_d}{dt} \right| \tag{6}$$

Where $U_d^t$ is the energy dissipation of a strain ε at any time t, J/cm³; $U_e^t$ is the elastic energy of a strain ε at any time t, J/cm³.

**2.2.2 Local surface deformation.** Local surface deformation of red sandstone corresponds to the process of energy accumulation and release. When external stress or internal strain energy exceeds the local strength of red sandstone, energy dissipates through plastic deformation or fracture, and the surface of red sandstone responds to the process of deformation. Surface images are collected synchronously by multiple cameras, and subset displacement tracking using the ZNSSD algorithm generates a 3D displacement field. To address isotropic deviations, the surface under investigation is divided into four symmetrical regions for collaborative analysis aimed at uncovering the patterns of deformation diffusion along the radial and axial directions., This approach enables the generation of a three-dimensional displacement map for the entire surface. Specifically, the sample's surface is divided into four symmetrical arc-shaped regions (C1-C4), corresponding to the displacement cloud map partitions DC1-DC4 (Fig 4). Therefore, the surface deformation can be calculated using the extracted data:

$$S_D = \sqrt[3]{OD^2 + HD^2 + VD^2} \tag{7}$$

Where $S_D$ is local surface deformation (mm), OD is out-of-plane displacement (mm), HD is horizontal displacement (mm), VD is vertical displacement (mm).

### 2.3 Correlation between energy and surface deformation

During the uniaxial loading of red sandstone, the surface deformation is influenced by the external load and the associated energy input, leading to corresponding alterations. Surface deformation serves as an external indication of energy input. Energy modifies the internal and external structure of red sandstone through thermodynamic or mechanical interactions, altering the surface morphology of red sandstone. According to the plastic mechanics theory, equation. (8) can be applied to rocks under complex stress state to represent the rela tionship between stress intensity and strain intensity [34]. Combining equations. (3) and (4), the relationship between energy and stress, the heat can be ignored in the loading. Therefore, the correlation is established as follows:

$$\sigma_i = \varphi(\varepsilon_i) = A\varepsilon_i^m \tag{8}$$

Where $\varepsilon_{SD}$ is the local deformation strain of the surface, and U is the input energy.

This study investigates the dynamic evolution of local red sandstone deformation and Poisson 's ratio from three aspects: sample preparation, testing, and data analysis. Firstly, samples undergo speckle processing before uniaxial compression testing. Stress-strain and surface deformation data are obtained using a full-time high-temporal-resolution 3D visualization test system for multi-field coupling damage analysis. Through data processing and analysis, the relationship between surface deformation and energy evolution of red sandstone is established.

## 3 Result and discussion

### 3.1 Test results

The deformation characteristics, axial strain-stress relationship, and surface deformation displacement of red sandstone under uniaxial compression are analyzed using a high spatial-temporal resolution 3D visualization test system for multi-field coupling damage.

**3.1.1 Deformation and failure characteristics of red sandstone.** The transition from ductile to brittle behavior in red sandstone samples CJ-1 to CJ-6 is determined based on the duration of the plastic deformation stage. The primary surface damage in samples CJ-1 to CJ-3 is concentrated at the bottom (highlighted within the yellow dotted box), whereas for CJ-4 to CJ-6, the damage zone is circled by a white ellipse. The number of obvious main cracks and secondary cracks is counted by Fig 6. CJ-1 and CJ-2 are dominated by 1–2 main cracks, CJ-5 and CJ-6 may have more cracks due to complex damage, CJ-3 and CJ-4 are in between. These reflect the transformation of failure mode from single crack propagation to multi-crack interweaving.

**3.1.2 Elastic energy and dissipative energy evolution.** Using equations (1) to (4), the energy evolution of six red sandstone specimens with different lithologies during uniaxial compression is calculated.

The energy dissipation process in fractured red sandstone under uniaxial compression follows distinct stages: nonlinear increase, linear increase, stabilization, mutation, and sharp rise. The peak strength of CJ-1∼CJ-2 is 8.02∼10.08 MPa, and the corresponding strain is 5.37∼6.14. The peak strength of CJ-3∼CJ-4 is 37.95∼38.47 MPa, and the corresponding strain is 6.98∼8.17. The peak strength of CJ-5∼CJ-6 is 44.75∼57.18 MPa, and the corresponding strain is 7.51∼11.28. This study's stage division, based on the stress-strain curve, aligns with previous findings. The peak ratio of dissipated energy to elastic energy density $K_{d/e}$ occurs primarily in stage I and II before declining. In Fig 8, the ratio of dissipation

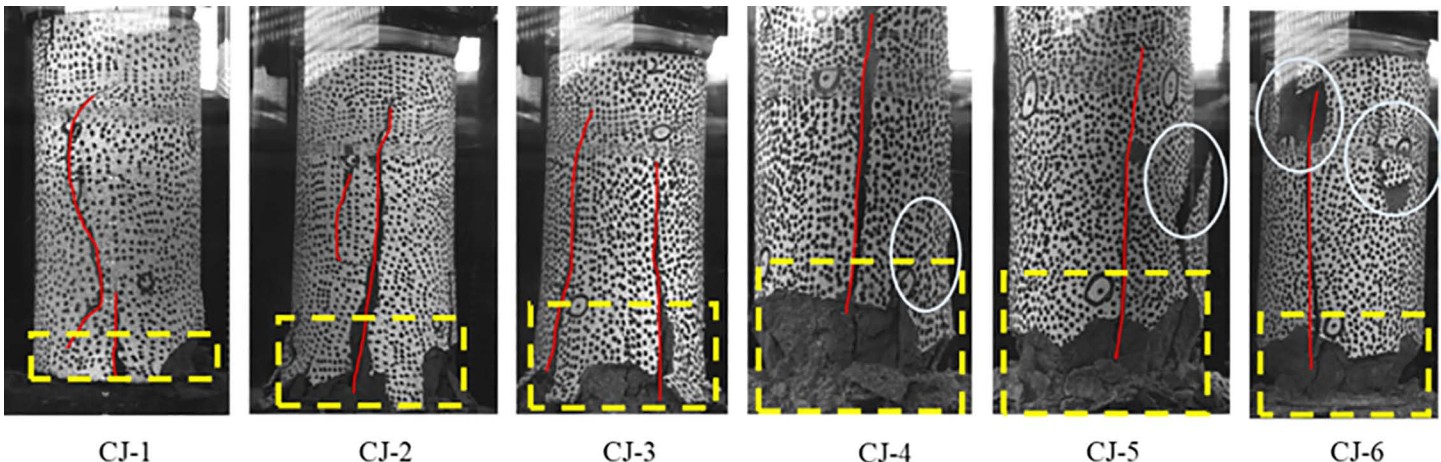

**Fig 6. Failure Modes.**

energy to elastic energy of CJ-1~CJ-2 is 25~45, the ratio of dissipation energy to elastic energy of CJ-3~CJ-4 is 55~110, and the ratio of dissipation energy to elastic energy of CJ-5~CJ-6 is 130~150. The relationship between the value of dissipated energy and elastic energy ratio and the sample is gradually increasing, which is the same as the peak value of the sample, as shown in Fig 8. Therefore, it can be concluded that the energy conversion of the sample in the failure stage increases with the increase of the peak intensity, indicating that the higher the strength of the rock sample, the more sufficient and uniform the energy flow in the sample, the more dispersed the distribution of the failure phenomenon in the sample, which corresponds to the gradual upward movement and expansion of the failure area of Fig 6 CJ-1~CJ-6. In general, the dissipated energy $U_d$ exceeds the releasable elastic strain energy $U_e$, indicating that most external energy is converted into surface energy and dissipates during rock sample compression. Under constant axial deformation rates, the rapid deformation of the rock side in the S1 and S2 stages in Fig 9 corresponds to this energy dissipation mechanism, rather than being stored as elastic energy $U_e$, with strain softening dominating until specimen failure in the loading process. With a similar mutation, the dissipation rate increases sharply as damage accumulates, exhibiting exponential growth until the sample undergoes total failure. In the initial stages (I and II), the dissipation energy rate remains low, with damage dispersed across multiple regions of the rock sample. With continuous loading, isolated damage zones coalesce into larger damage regions. As evident from Fig 7, this phase corresponds to stage III, which subsequently progresses into a state of damage.

**3.1.3 Variation law of surface displacement.** By utilizing the acquired deformation failure, stress-strain curve, and surface deformation displacement data, a more in-depth analysis of the failure characteristics of red sandstone

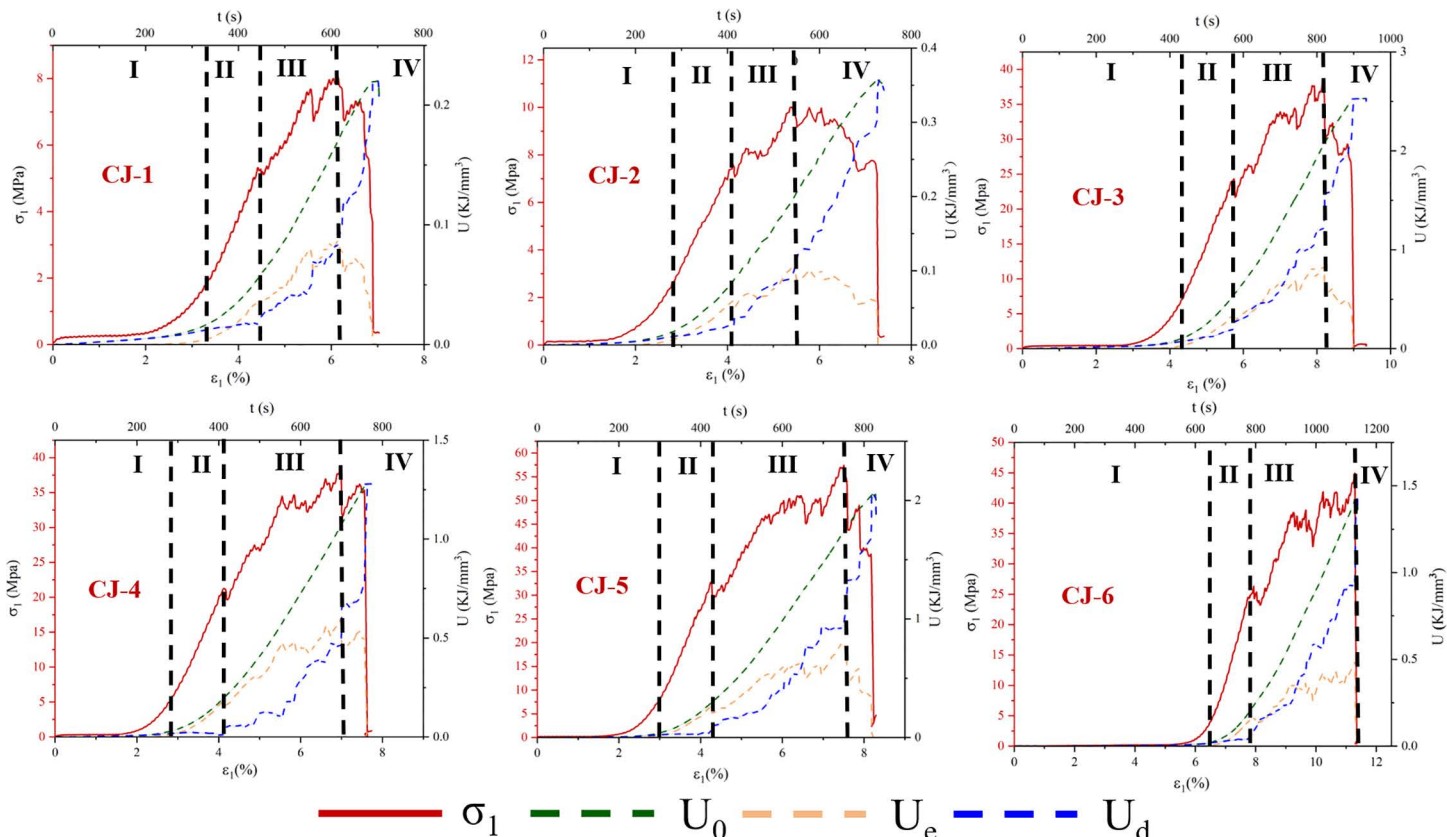

**Fig 7. Sample Stress-Strain and Energy Evolution.**

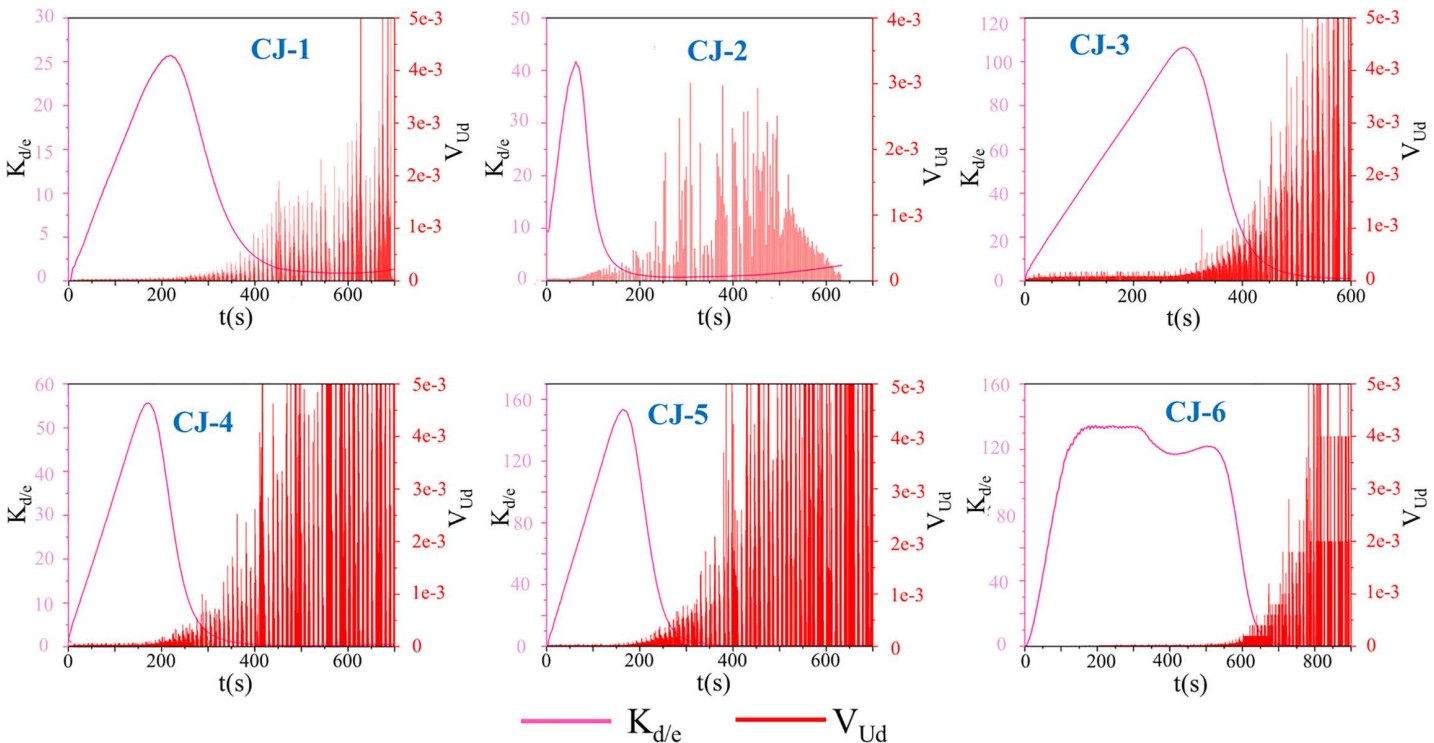

**Fig 8. Ratio of Dissipated Energy to Elastic Energy and Dissipation Rate.**

transitioning from ductile to brittle rock failure under uniaxial compression are conducted. The evolution of elastic energy and dissipative energy, along with their relationships, as well as the variation patterns of surface displacement, are analyzed and categorized.

The sample's surface displacement shows slow rise, sharp rise, rise-nearly gentle/ decline, and rise four states throughout the test, these four stages are recorded as S1, S2, S3 and S4 respectively. Combined with Fig 9 and Table 3, the lateral displacement stages S1 and S2 corresponding to stage I-initial compaction stage, the lateral displacement stage S3 corresponding stage II-elastic stage and III-unstable fracture development, and the lateral displacement S4 corresponding stage IV-failure. In the stage S1 and S2, the lateral displacement changes rapidly, indicating that the rock expands outward sharply at this time. The change rate of the S1 stage is smaller than that of the S2 stage. The S1 stage can be defined as the initial compaction stage of the lateral displacement, which is mainly filled with the gap in the horizontal direction, and the particles move in the radial direction; the rate of change in the S2 stage is large, because in the S1 stage, the gap in the horizontal direction has been largely filled, so the horizontal deformation expansion is significant at this stage. The change of S3 stage is relatively gentle. Combined with the III-unstable fracture development stage of Fig 7, the axial stress fluctuates, and there is no obvious deformation and failure on the surface. This is because the internal collapse of the rock occurs in the interior, which corresponds to the decrease of the lateral displacement. The displacement of the S4 stage generally rises, and there is also a decrease. Combined with Fig 7, the rock has been obviously damaged at this time. According to the failure mode of Fig 6, the part of the curve falling is the area where the rock fragments fall off and the depression appears. The rising area is the area where the rock expands and fails to fall off.

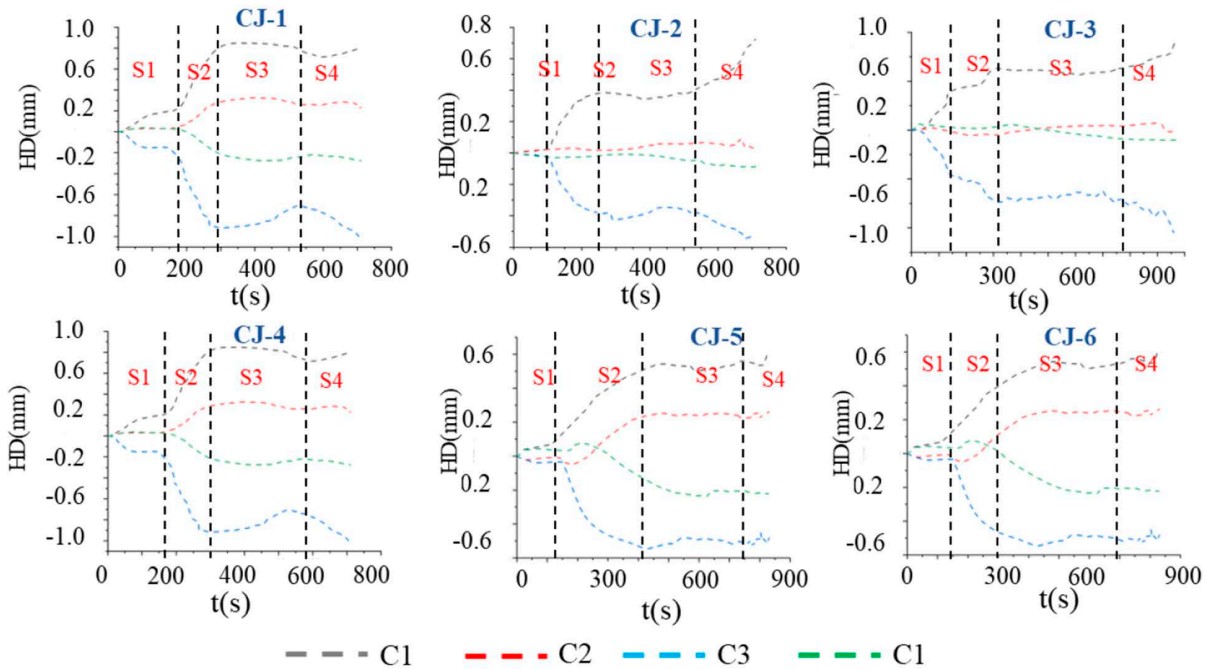

（a）Horizontal Displacement

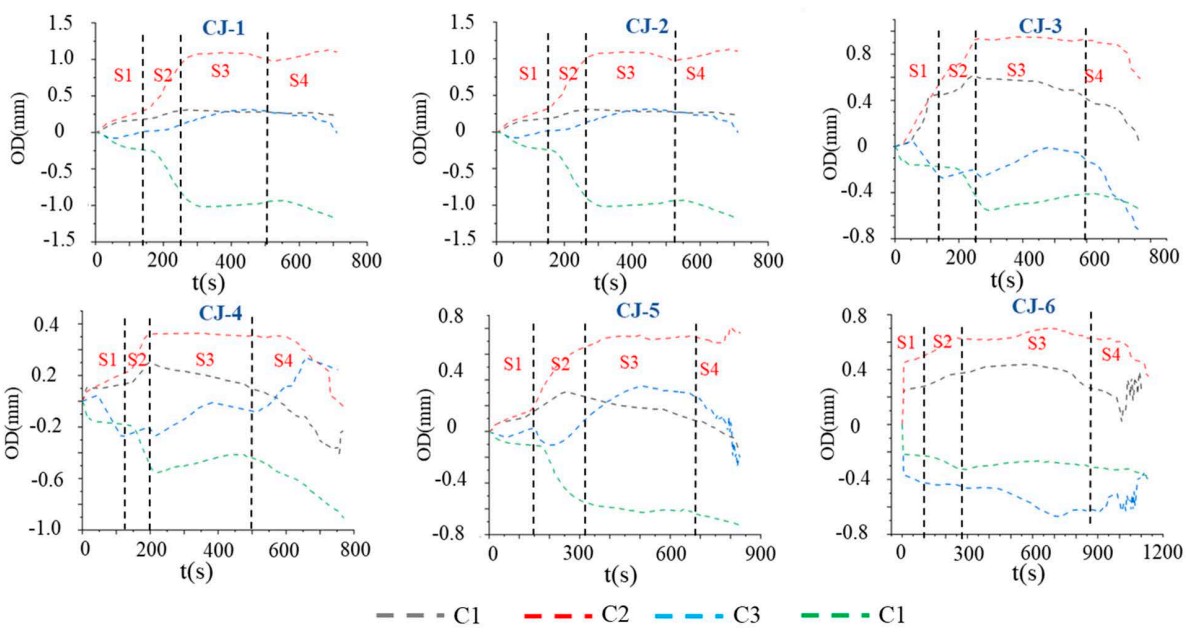

（b）Longitudinal Displacement

**Fig 9. Regional Horizontal and Out-of-Plane Displacement. (a)** Horizontal Displacement. **(b)** Longitudinal Displacement.

**Table 3. Data Statistics and Analysis.**

| | $\sigma_c$(MPa) | $\varepsilon_{1c}$(%) | | S1 | S2 | S3 | S4 | I | II | III | IV |
|---|---|---|---|---|---|---|---|---|---|---|---|
| CJ-1 | 8.02 | 6.14 | $\varepsilon 1$(%) | | | | | 0-3.2 | 3.2-4.3 | 4.3-6.14 | 6.14- |
| | | | t(s) | 0-185 | 185-296 | 296-542 | 542- | 0-320 | 320-430 | 430-614 | 614- |
| CJ-2 | 10.08 | 5.37 | $\varepsilon 1$(%) | | | | | 0-2.85 | 2.85-4.08 | 4.08-5.37 | 5.37- |
| | | | t(s) | 0-102 | 102-250 | 250-545 | 545- | 0-285 | 285-408 | 408-537 | 537- |
| CJ-3 | 37.95 | 8.17 | $\varepsilon 1$(%) | | | | | 0-4.2 | 4.2-5.86 | 5.86-8.17 | 8.17- |
| | | | t(s) | 0-148 | 148-329 | 329-785 | 785- | 0-420 | 420-586 | 586-817 | 817- |
| CJ-4 | 38.47 | 6.98 | $\varepsilon 1$(%) | | | | | 0-2.95 | 2.95-4.12 | 4.12-6.98 | 6.98- |
| | | | t(s) | 0-162 | 162-300 | 300-586 | 586- | 0-295 | 295-412 | 412-698 | 698- |
| CJ-5 | 57.18 | 7.51 | $\varepsilon 1$(%) | | | | | 0-2.96 | 2.96-4.25 | 4.25-7.51 | 7.51- |
| | | | t(s) | 0-126 | 126-406 | 406-749 | 749- | 0-296 | 296-425 | 425-751 | 751- |
| CJ-6 | 44.75 | 11.28 | $\varepsilon 1$(%) | | | | | 0-6.32 | 6.32-7.95 | 7.95-11.28 | 11.28- |
| | | | t(s) | 0-148 | 148-261 | 261-762 | 762- | 0-623 | 623-795 | 795-1128 | 1128- |

Note: $\sigma_c$ represents peak strength, $\varepsilon_{1c}$ corresponds to axial strain at peak strength, and S1-S4 and I-IV denote regional surface displacement and stress-strain stage divisions, respectively.

## 3.2 Correlation between energy evolution and surface deformation

The correlation between energy input and Poisson's ratio in red sandstone is mainly reflected in the process of energy-driven internal damage evolution. As external energy input increases, elastic strain energy gradually accumulates inside the red sandstone, leading to microcrack propagation and an increase in transverse expansion and Poisson's ratio. Especially in the post-peak stage, crack propagation and frictional slip cause extensive energy dissipation, further increasing Poisson's ratio (Fig 10).

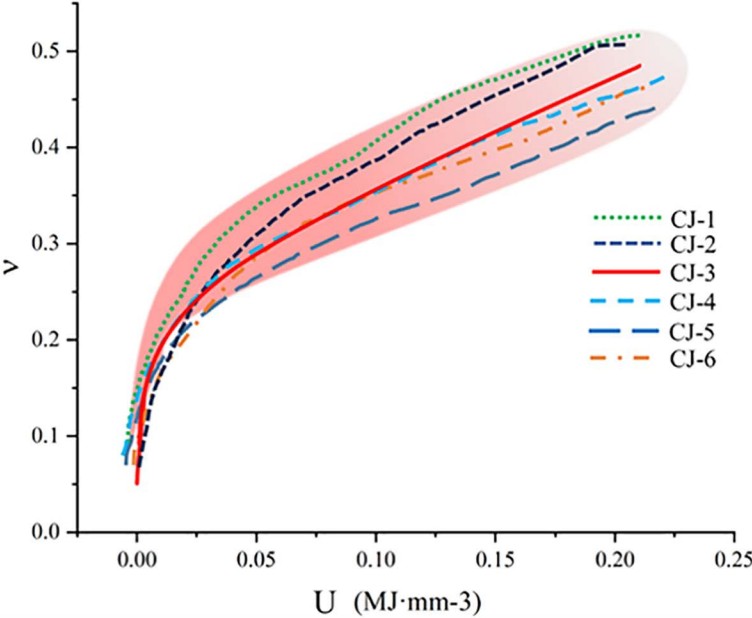

**Fig 10. Relationship Between Energy and Poisson 's ratio.**

Fig 10 demonstrates that different red sandstone samples exhibit a consistent trend in Poisson's ratio and energy. By averaging Poisson 's ratio and energy values across multiple specimens, a general trend curve can be derived to describe the relationship between Poisson's ratio and energy of different samples mathematically, as follows:

$$U = -0.01686 + 0.0058 \times e^{(7.58582*v)} \tag{9}$$

The eight-camera system (Fig 2b) captures standard rock samples from four angles, assembling a complete view of the red sandstone surface using four surfaces taken. The above deformation characteristics of red sandstone reveal that failure initiates at the bottom of the red sandstone specimen, with deformation occurring earlier in the lower region compared to the middle and upper sections. Therefore, the surface deformation displacement data are extracted from the bottom regions (C1-b~C4-b) of the four captured surfaces. Utilizing this data, alongside the external input energy, the following diagram is drawn:

Drawing upon the above-mentioned data pertaining to energy input and surface deformation displacement of red sandstone, Fig 11 has been drawn. This figure elucidates the growth trends observed in samples CJ-2~CJ-6, which initially exhibit slow growth before transitioning to a phase of rapid increase. In contrast, sample CJ-1 demonstrates a nearly linear growth pattern. The divergence in these growth behaviors can be attributed to the low porosity and strength of the sample, which facilitates a consistent and steady accumulation of deformation energy during the compression process.

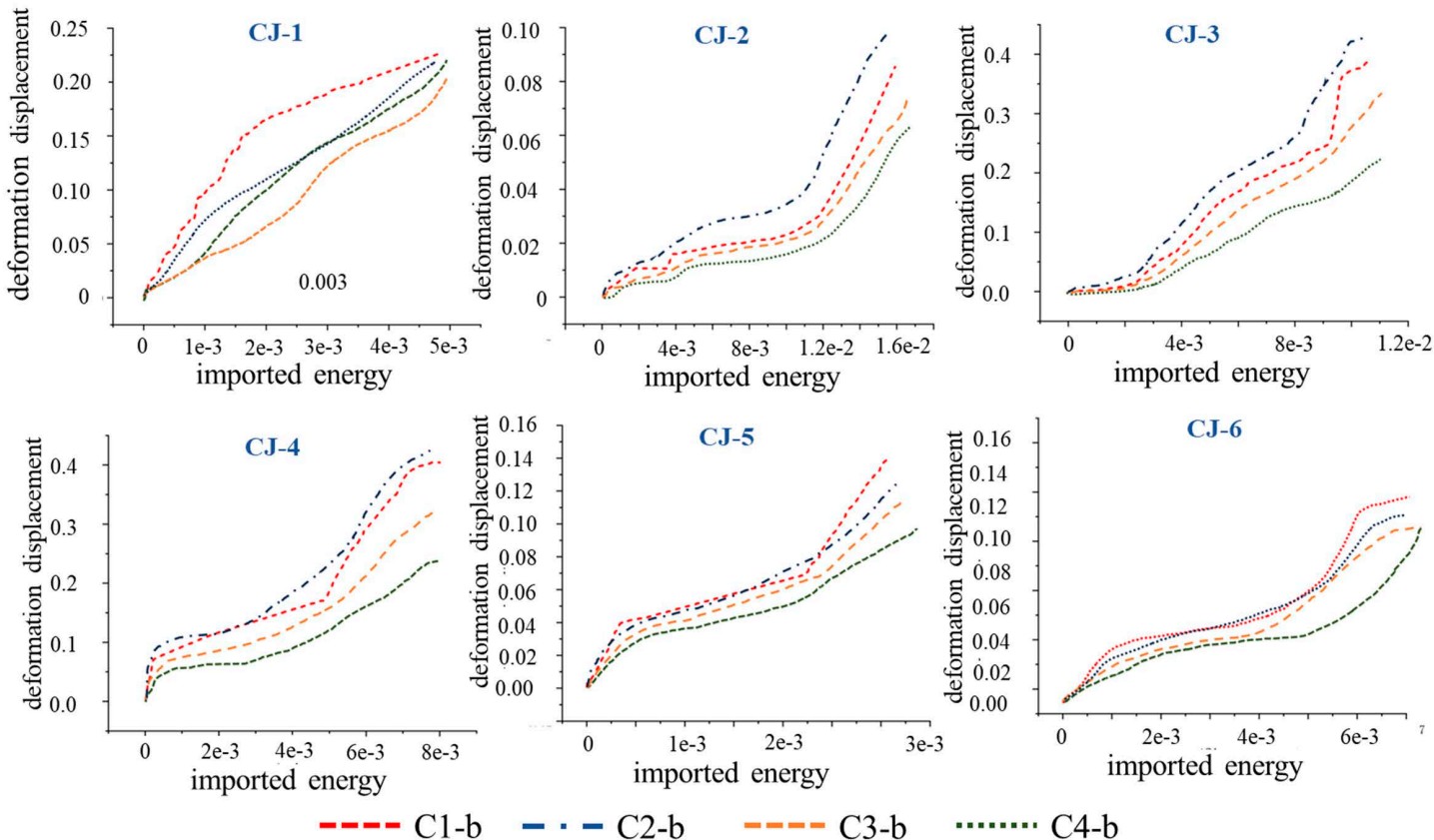

**Fig 11. Energy Input vs. Surface Displacement.**

The above data sets are plotted within the same coordinate system, with the data range demarcated by the pink area in the following picture. The intensity of color indicates the degree of data overlap in the area, wherein a deeper hue signifies a greater degree of consistency. It is obvious that the trend exhibited by the data within this region aligns closely with the curve trend in Fig 11. Therefore, by computing the average value of the data points within the specified range, a trend line can be obtained, as shown in Fig 12 (a) below. The trend line is fitted as shown in Fig 12 (b).

According to the fitting curve and the original trend curve, it is evident that in the early stage, the curve exhibits sharp variations, while the energy input is low and the fitting goodness is inadequate. When considering the actual situation, at this stage, the external energy input or applied load is relatively minor, and the deformation is negligible. Consequently, the significance of the early stage, in terms of its contribution to the overall behavior, is minimal and can be disregarded. According to the above results, the mathematical relationship between external energy input and local deformation can be articulated as follows:

$$\varepsilon_{SD} = S_D/S_d \tag{10}$$

$$\varepsilon_{SD} = -2.15E04 * e^{-e(-7.5(E-0.3))} \tag{11}$$

The deformation process of rock under external load is the evolution of elastic energy and dissipated energy. The input energy is mainly dissipated energy in the deformation and failure of the surface. Therefore, the relationship between local deformation and dissipated energy on the surface can better reflect the evolution relationship between deformation and energy in the process of rock failure.

It can be seen from Fig 13 that the dissipation energy and input energy of different samples have similar trends, but their values are different. Therefore, the value is enlarged to the value of CJ-3 (the largest value) sample by processing. Therefore, the following Fig 14 is obtained:

According to the results, the mathematical relationship between dissipated energy and local deformation can be articulated as follows

$$U_d = -0.2 + 0.24 * exp(0.95 * U) \tag{12}$$

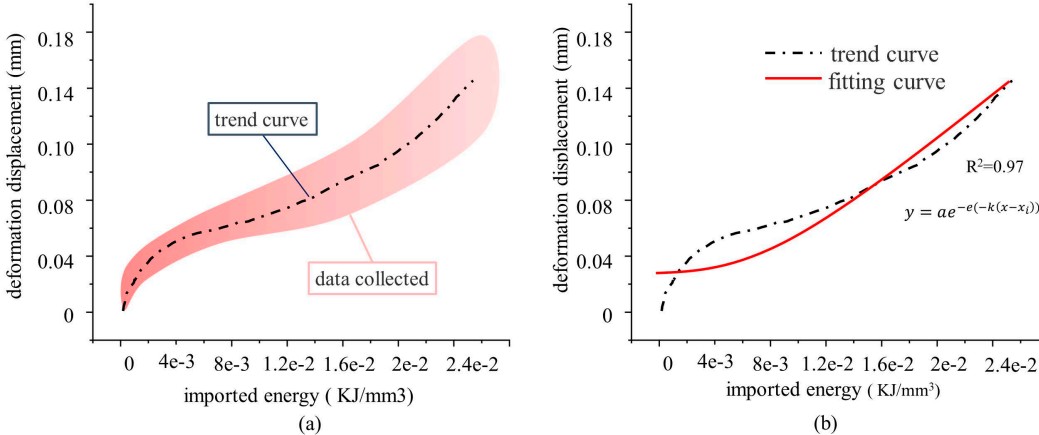

**Fig 12. Relationship Between Energy Input and Surface Deformation Displacement.**

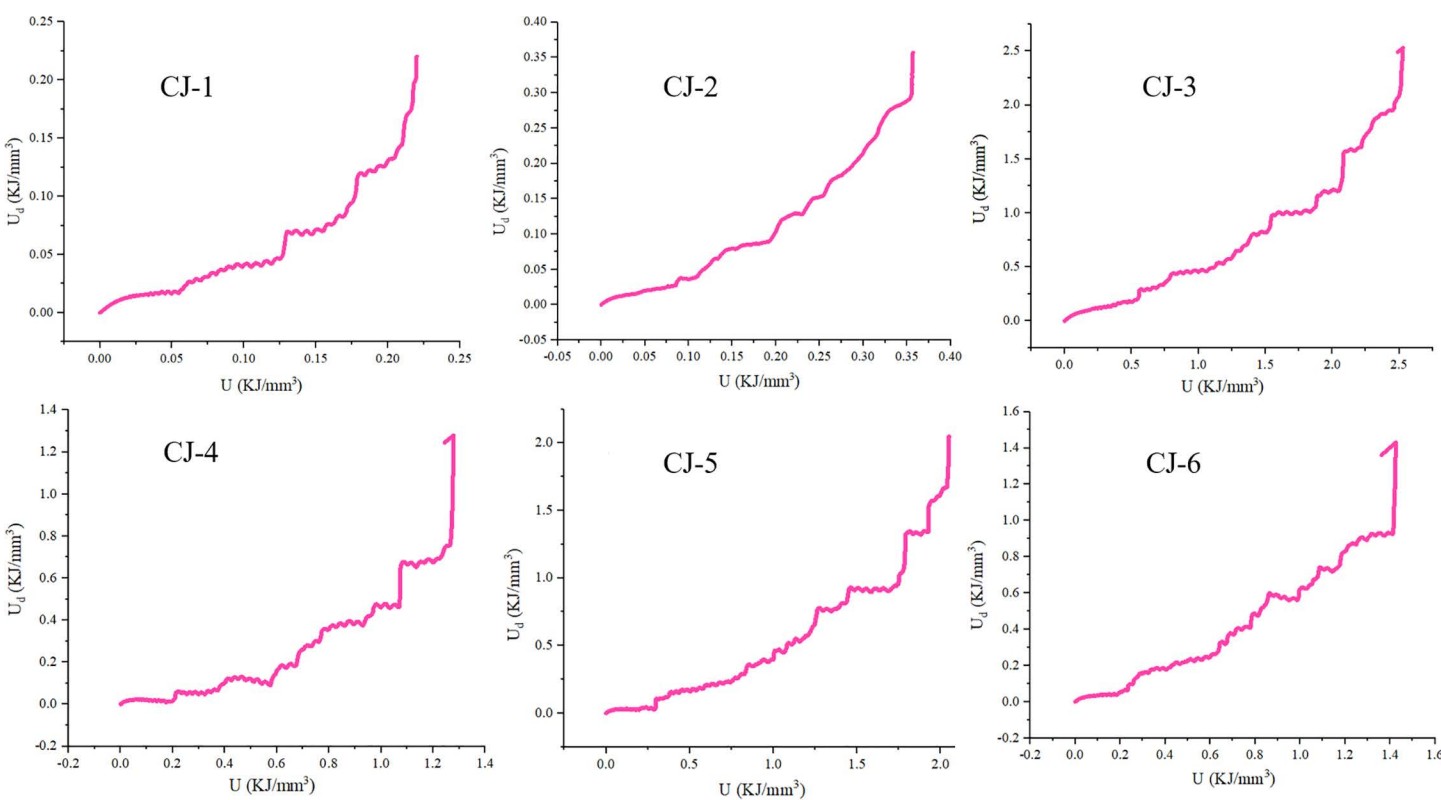

**Fig 13. The relationship between dissipated energy $U_d$ and total input energy U.**

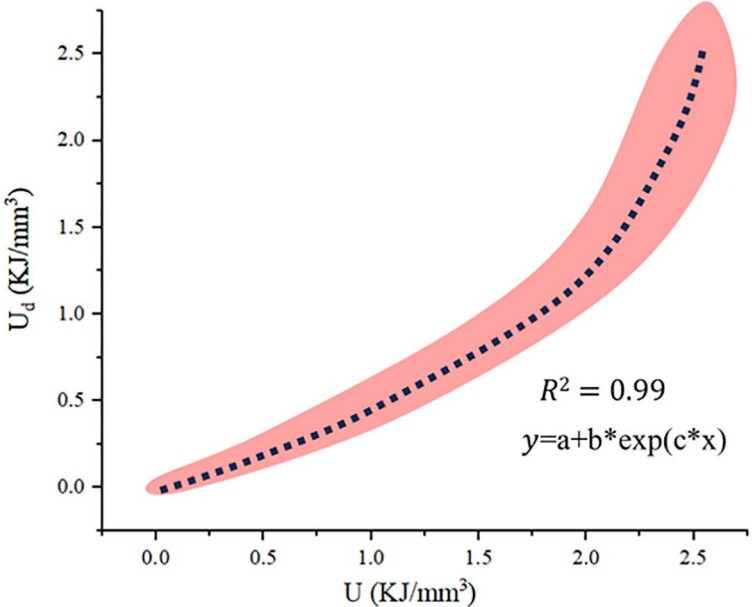

$$R^2 = 0.99$$
$$y = a + b*\exp(c*x)$$

**Fig 14. The fitting relationship between dissipated energy and input energy.**

$$U = In\left(\frac{U_d + 0.2}{0.24}\right)/0.95 \qquad (13)$$

Combining (10) with (11) and (12):

$$\varepsilon_{SD} = -2.15E04 * e^{-e\left(-7.5\left(\ln\left(\frac{U_d + 0.2}{0.24}\right)/0.95 - 0.3\right)\right)} \qquad (14)$$

Where $\varepsilon_{SD}$ is the local deformation strain of the surface, $s_d$ is the sample diameter (mm), $U_d$ is the dissipated energy and U is the input energy (MJ· mm-3).

### 3.3 Discussion

The above results indicate a correlation between local deformation and energy on the surface of red sandstone. This study, based on the results of uniaxial compression testing and a high spatial-temporal resolution 3D visualization test system for multi-field coupling damage, explores the relationship between local deformation and energy evolution in red sandstone. The derived correlation is expressed in Equation (11).

The surface displacement evolution of red sandstone is classified into four stages: S1~S4. In Stage S1 (gentle deformation), internal radial voids are filled, and surface cracks form under axial compression, leading to slow deformation growth. Stage S2 (rapid deformation) is characterized by crack penetration and void filling, resulting in a sharp increase in displacement. In Stage S3 (stable deformation), the upward expansion at the bottom facilitates energy dissipation through the collapse of surface cracks and internal microcracks, causing displacement fluctuations to stabilize. Stage S4 (disordered deformation) is marked by rock detachment and erratic displacement increments. There exists a time sequence disparity between the evolution process and the axial stress-strain. Stages S1~S2 correspond to the compaction phase (Stage I), while Stage S3 spans the elastic deformation phase (Stage II) and unstable fracture phase (Stage III). Stage S4 aligns with the failure phase (Stage IV). The cross-scale response mechanism reflects the lag of lateral deformation relative to axial failure. Failure mode analysis (Fig 6) reveals that plastic rock failure is concentrated at the bottom due to stress concentration and plastic deformation accumulation caused by bottom uplift loading. Due to the high brittleness and robust surface confinement, brittle rocks tend to accumulate stress in the upper part, resulting in an upward migration of the failure zone. The stringent surface constraints of brittle rocks restrict their deformation, localizing the deformation process. When the load increases, the stress is abruptly released, precipitating failure in the upper half. In contrast, the damage at the bottom of plastic rocks does not propagate upward following the damage. This difference reveals the divergent failure pathways influenced by the interplay between the mechanical properties of the rock mass and the loading mode, providing a theoretical basis for evaluating red sandstone stability in engineering applications. The relationship between energy and local surface deformation is illustrated in Fig 12 (b). In the early stages, deformation changes rapidly with energy input, albeit at a small magnitude. This is because deformation in Stage S1 primarily results from surface crack propagation, which can be approximately generated from scratch. From the above analysis, it is evident during this stage, the primary deformation in red sandstone occurs internally. Consequently, while the rate of change is rapid, the magnitude of deformation remains relatively small, and the deformation process appears gradual when viewed on a temporal scale. The relationship between the surface deformation and the input energy of the red sandstone in the mid-to-late stages follows an approximately exponential function, with deformation increasing exponentially as energy accumulates. This behavior arises because, as energy is input, rock porosity gradually decreases, and rock particles gradually diffuse outward. As the internal voids are gradually filled, outward diffusion intensifies, leading to this phenomenon and exponential surface deformation until failure occurs. Given the minor deformation observed in the early stage, its proportional contribution to the overall deformation is negligible. Therefore, the relationship between energy input and surface deformation in red sandstone can be approximated as an exponential function (Equation 11). Surface displacement experiences a sharp increase with continued energy input.

Throughout the surface deformation process of red sandstone, deformation does not exhibit a stable and continuous growth, but increases sharply in response to energy input. This sharp increase also indicates the abrupt nature of deformation and failure. Therefore, establishing a deformation-energy evolution correlation of red sandstone allows for the prediction of surface deformation magnitude and extent of red sandstone under external energy input.

## 4 Conclusion

(1) Based on uniaxial compression testing, the energy evolution and surface deformation of red sandstone under uniaxial compression are investigated. The surface deformation of red sandstone is classified in terms of its magnitude and progression, while considering the energy evolution of red sandstone. The relationship between the input energy, dissipated energy and surface deformation of red sandstone is established, addressing the unclear correlation mechanism governing surface deformation of red sandstone under loading or other energy inputs.

(2) In this study, the surface deformation of red sandstone is divided into four stages (S1, S2, S, S4) by compressing red sandstone, which corresponds to the axial deformation stage but the time is not synchronized. The energy characteristics of each stage reveal that while total energy input remains low in S1 and S2, substantial deformation occurs due to the breakdown of the initial stability of the red sandstone surface. In S3, energy input increases more slowly, but accumulation continues in general, leading to stable growth in this stage. At the end of this stage, as energy input persists, deformation begins to increase sharply. In S4, energy input continues to increase, due to the deformation accumulation occurring at the S1~S3 stage, the red sandstone in this stage finally experiences deformation and failure. The surface becomes convex or concave, causing irregular deformation, which reveals the evolution mechanism of non-uniform distribution of local deformation displacement in red sandstone under the overall action of compressive load.

(3) This study establishes an energy evolution-surface deformation correlation model for red sandstone. The findings have practical implications for engineering applications such as dams, and slopes, where red sandstone deformation data can inform the design of support structures. Real-time deformation monitoring can aid in identifying potential dangerous areas in advance and serve as a reference for the deformation characteristics and behavior of rocks in other projects involving free surfaces.

## Supporting information

**S1 Data.** Data set.
(ZIP)

## Author contributions

**Data curation:** Feng Gao, Guangjun Cui.

**Formal analysis:** Feng Gao, Guangjun Cui, Jin Liao, Zhen Liu.

**Funding acquisition:** Zhen Liu, Cuiying Zhou.

**Investigation:** Feng Gao, Guangjun Cui, Chunhui Lan, Ziyu Tao.

**Methodology:** Feng Gao, Guangjun Cui, Jin Liao, Zhen Liu, Chunhui Lan, Ziyu Tao.

**Writing – original draft:** Feng Gao, Jin Liao.

**Writing – review & editing:** Feng Gao, Zhen Liu, Cuiying Zhou.

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
