## [Decision Letter · Decision Letter 0]

17 Jun 2025

PONE-D-25-28363Relationship between Surface Deformation Displacement and Energy Evolution of Red Sandstone under Uniaxial CompressionPLOS ONE

Dear Dr. Liu,

Thank you for submitting your manuscript to PLOS ONE. After careful consideration, we feel that it has merit but does not fully meet PLOS ONE’s publication criteria as it currently stands. Therefore, we invite you to submit a revised version of the manuscript that addresses the points raised during the review process.

We look forward to receiving your revised manuscript.

Kind regards,

Lisong Zhang

Academic Editor

PLOS ONE

Journal Requirements:

“Funding The research is supported by the National Natural Science Foundation of China (NSFC) (Grant No. 42293354, 42277131, 42293351, 42293355, 42293350). “

“The work presented in this article was supported by the National Natural Science Foundation of China (NSFC) (Grant No. 42293354, 42277131, 42293351, 42293355, 42293350).”

“Funding The research is supported by the National Natural Science Foundation of China (NSFC) (Grant No. 42293354, 42277131, 42293351, 42293355, 42293350). “

5. Please ensure that you refer to Figure 1, 5 and 8 in your text as, if accepted, production will need this reference to link the reader to the figure.

6. Please upload a copy of Figure 14, to which you refer in your text on page 35 in PDF submission. If the figure is no longer to be included as part of the submission please remove all reference to it within the text.

7. We note you have included a table to which you do not refer in the text of your manuscript. Please ensure that you refer to Table 3 in your text; if accepted, production will need this reference to link the reader to the Table.

**Additional Editor Comments:**

Reviewer 1

The manuscript "Relationship between Surface Deformation Displacement and Energy Evolution of Red Sandstone under Uniaxial Compression" provides an accurate analysis of the correlation between surface deformation and energy in red sandstone under loading, revealing the surface deformation process of red sandstone. Based on deformation magnitude and degree, the study identifies four distinct surface deformation stages that correspond to axial deformation stages but exhibit temporal discrepancies, offering important theoretical and technical support for deformation control in free-surface engineering projects such as slopes. However, there remain some minor issues in the manuscript that require revision. Overall, I recommend minor revisions.

General opinion

1. Regarding the description of previous unstudied aspects in lines 69-72 within Chapter 2, should this be moved to the Introduction?

2. There are numerous formatting issues in the text. Please carefully review and revise them, for example:

Both the figures in Line 383 and Line 388 are labeled as "Fig. 13.";

There is no blank line after Line 383 and Line 388, with text directly following. In contrast, other figures and paragraphs have blank lines in between;

Some parts use "Fig. xx," while others use the full term "Figure," and some are written as "Fig.xx" (without a space);

The format of the title in Line 124, "Fig.1 Test Flow Chart," should be consistent with other figures;

Citation formats in the article should be standardized and unified;

3. In Section 3.1.3, where is the analysis of surface displacement changes based on the charts?

4. In Fig. 11, some subplots have labeled x-axes, while others are missing labels.

5. Conclusion (2) is overly complex. Please refine it by extracting the key points.

Reviewer 2

This paper entitled " Relationship between Surface Deformation Displacement and Energy Evolution of Red Sandstone under Uniaxial Compression" presents investigates the relationship between surface deformation and energy evolution in red sandstone under uniaxial compression using advanced 3D visualization techniques. It is an interesting topic that may inspire the research in rock mechanics. While the topic is relevant to geotechnical engineering, several significant issues need to be addressed.

1. In line 21, the "red rock sandstone" should be "red sandstone" (redundant terminology).

2. In line 27-28, the sentence “According to the amount and degree of deformation” is not clear, please clarify.

3. In the Introduction section, literature review appears to be somewhat limited in scope. Including more recent and diverse studies could provide a more comprehensive background. Please consider the following papers: 10.1007/s00603-022-02861-4, and 10.1007/s00603-024-03845-2.

4. Please improve the state in the manuscript such as “Fig.1 Test Flow Chart” and “Research Content and Methods”.

5. In lines 174-189, sample preparation description lacks important details about specimen uniformity and quality control

6. In Section 2.3, the correlation between energy and surface deformation demonstrated in Equation 8 lacks theoretical foundation, please give more details explanation.

7. In lines 268-271, failure mode classification lacks quantitative criteria and it seems that the failure modes can be distinguished from Fig. 6.

8. The manuscript contains several grammatical errors and awkward sentence structures that need to be addressed. A thorough proofreading is recommended to improve readability.

Reviewer 3

This paper focuses on the energy evolution process of surface deformation of red sandstone under uniaxial compression, and uses a 3D visualization testing system to observe surface deformation and conduct correlation analysis with the energy changes during the failure process of red sandstone. The manuscript is well written and the topic is interesting and novel. However, some problems were also found in the manuscript. Therefore, minor revisions are required before the paper is published in the journal. All details are summarized below.

Required changes:

Point 1: The sentence, “The energy input resulting from construction disturbance (such as tunneling and pile foundation construction) inevitably induces deformation and failure, potentially compromising project safety”, mentioned in the Introduction, are reference for these statement and/or previous works? Please add relevant references.

Point 2: The descriptions of sample composition in Tables 1 and 2 are irrelevant to the research content of this paper, which focuses on external energy input and surface deformation. It is recommended to remove them.

Point 3: In the Fig. 6 Failure Modes, merely outlines the failure phenomena without providing concrete explanations. It is recommended to provide a more in-depth explanation of the failure modes observed in different samples, along with a thorough analysis of the underlying mechanisms responsible for these phenomena.

Point 4: The axis labels in several figures appear pixelated and difficult to read. The authors should provide higher-resolution versions with clearly legible text.

Point 5: The paper contains formatting and content inaccuracies that require correction. It is recommended to further improve them. It is necessary to correct the whole paper for accuracy and precision. Minor grammatical problems in the full text need to be corrected.

Reviewers' comments:

Reviewer's Responses to Questions

**Comments to the Author**

1. Is the manuscript technically sound, and do the data support the conclusions?

Reviewer #1: Yes

Reviewer #2: Yes

Reviewer #3: Yes

2. Has the statistical analysis been performed appropriately and rigorously? 

Reviewer #1: Yes

Reviewer #2: Yes

Reviewer #3: Yes

3. Have the authors made all data underlying the findings in their manuscript fully available?

Reviewer #1: Yes

Reviewer #2: Yes

Reviewer #3: Yes

4. Is the manuscript presented in an intelligible fashion and written in standard English?

Reviewer #1: Yes

Reviewer #2: Yes

Reviewer #3: Yes

5. Review Comments to the Author

Reviewer #1: Review comments

The manuscript "Relationship between Surface Deformation Displacement and Energy Evolution of Red Sandstone under Uniaxial Compression" provides an accurate analysis of the correlation between surface deformation and energy in red sandstone under loading, revealing the surface deformation process of red sandstone. Based on deformation magnitude and degree, the study identifies four distinct surface deformation stages that correspond to axial deformation stages but exhibit temporal discrepancies, offering important theoretical and technical support for deformation control in free-surface engineering projects such as slopes. However, there remain some minor issues in the manuscript that require revision. Overall, I recommend minor revisions.

General opinion

1. Regarding the description of previous unstudied aspects in lines 69-72 within Chapter 2, should this be moved to the Introduction?

2. There are numerous formatting issues in the text. Please carefully review and revise them, for example:

Both the figures in Line 383 and Line 388 are labeled as "Fig. 13.";

There is no blank line after Line 383 and Line 388, with text directly following. In contrast, other figures and paragraphs have blank lines in between;

Some parts use "Fig. xx," while others use the full term "Figure," and some are written as "Fig.xx" (without a space);

The format of the title in Line 124, "Fig.1 Test Flow Chart," should be consistent with other figures;

Citation formats in the article should be standardized and unified;

3. In Section 3.1.3, where is the analysis of surface displacement changes based on the charts?

4. In Fig. 11, some subplots have labeled x-axes, while others are missing labels.

5. Conclusion (2) is overly complex. Please refine it by extracting the key points.

Reviewer #2: This paper entitled " Relationship between Surface Deformation Displacement and Energy Evolution of Red Sandstone under Uniaxial Compression" presents investigates the relationship between surface deformation and energy evolution in red sandstone under uniaxial compression using advanced 3D visualization techniques. It is an interesting topic that may inspire the research in rock mechanics. While the topic is relevant to geotechnical engineering, several significant issues need to be addressed.

1. In line 21, the "red rock sandstone" should be "red sandstone" (redundant terminology).

2. In line 27-28, the sentence “According to the amount and degree of deformation” is not clear, please clarify.

3. In the Introduction section, literature review appears to be somewhat limited in scope. Including more recent and diverse studies could provide a more comprehensive background. Please consider the following papers: 10.1007/s00603-022-02861-4, and 10.1007/s00603-024-03845-2.

4. Please improve the state in the manuscript such as “Fig.1 Test Flow Chart” and “Research Content and Methods”.

5. In lines 174-189, sample preparation description lacks important details about specimen uniformity and quality control

6. In Section 2.3, the correlation between energy and surface deformation demonstrated in Equation 8 lacks theoretical foundation, please give more details explanation.

7. In lines 268-271, failure mode classification lacks quantitative criteria and it seems that the failure modes can be distinguished from Fig. 6.

8. The manuscript contains several grammatical errors and awkward sentence structures that need to be addressed. A thorough proofreading is recommended to improve readability.

Reviewer #3: This paper focuses on the energy evolution process of surface deformation of red sandstone under uniaxial compression, and uses a 3D visualization testing system to observe surface deformation and conduct correlation analysis with the energy changes during the failure process of red sandstone. The manuscript is well written and the topic is interesting and novel. However, some problems were also found in the manuscript. Therefore, minor revisions are required before the paper is published in the journal. All details are summarized below.

Required changes:

Point 1: The sentence, “The energy input resulting from construction disturbance (such as tunneling and pile foundation construction) inevitably induces deformation and failure, potentially compromising project safety”, mentioned in the Introduction, are reference for these statement and/or previous works? Please add relevant references.

Point 2: The descriptions of sample composition in Tables 1 and 2 are irrelevant to the research content of this paper, which focuses on external energy input and surface deformation. It is recommended to remove them.

Point 3: In the Fig. 6 Failure Modes, merely outlines the failure phenomena without providing concrete explanations. It is recommended to provide a more in-depth explanation of the failure modes observed in different samples, along with a thorough analysis of the underlying mechanisms responsible for these phenomena.

Point 4: The axis labels in several figures appear pixelated and difficult to read. The authors should provide higher-resolution versions with clearly legible text.

Point 5: The paper contains formatting and content inaccuracies that require correction. It is recommended to further improve them. It is necessary to correct the whole paper for accuracy and precision. Minor grammatical problems in the full text need to be corrected.

6. PLOS authors have the option to publish the peer review history of their article (what does this mean? ). If published, this will include your full peer review and any attached files.

**Do you want your identity to be public for this peer review?** For information about this choice, including consent withdrawal, please see our Privacy Policy .

Reviewer #1: No

Reviewer #2: No

Reviewer #3: No

---

## [Author Response · Author response to Decision Letter 1]

2 Jul 2025

Response to Academic Editor

Dear Editors and Reviewers:

Thank you for your letter and for the reviewers’ comments concerning our manuscript entitled “Relationship between Surface Deformation Displacement and Energy Evolution of Red andstone under Uniaxial Compression” (ID: PONE-D-25-28363). Those comments are all valuable and very helpful for revising and improving our paper, as well as the important guiding significance to our researches. We have studied comments carefully and have made correction which we hope meet with approval.

Response: Thank you very much for your review of this article and your valuable comments. It has been modified according to the template

“Funding The research is supported by the National Natural Science Foundation of China (NSFC) (Grant No. 42293354, 42277131, 42293351, 42293355, 42293350). “

Response: Thank you very much for your review of this article and your valuable comments. The fund support in the article has been deleted

“The work presented in this article was supported by the National Natural Science Foundation of China (NSFC) (Grant No. 42293354, 42277131, 42293351, 42293355, 42293350).”

“Funding The research is supported by the National Natural Science Foundation of China (NSFC) (Grant No. 42293354, 42277131, 42293351, 42293355, 42293350). “

Response: Thank you very much for your review of this article and your valuable comments. The fund support in the article has been deleted

Response: Thank you very much for your review of this article and your valuable comments. Have added accessories.

5. Please ensure that you refer to Figure 1, 5 and 8 in your text as, if accepted, production will need this reference to link the reader to the figure.

Response: Thank you very much for your review of this article and your valuable comments. References to Figure 1, 5, 8 have been added to the text.

6. Please upload a copy of Figure 14, to which you refer in your text on page 35 in PDF submission. If the figure is no longer to be included as part of the submission please remove all reference to it within the text.

Response: Thank you very much for your review of this article and your valuable comments. After modifying the serial number of the picture, the serial number has been increased to 14 pictures.

7. We note you have included a table to which you do not refer in the text of your manuscript. Please ensure that you refer to Table 3 in your text; if accepted, production will need this reference to link the reader to the Table.

Response: Thank you very much for your review of this article and your valuable comments. Table 3 has been added to the new content.

Response to Reviewer

Dear Editors and Reviewers:

Thank you for your letter and for the reviewers’ comments concerning our manuscript entitled “Relationship between Surface Deformation Displacement and Energy Evolution of Red andstone under Uniaxial Compression” (ID: PONE-D-25-28363). Those comments are all valuable and very helpful for revising and improving our paper, as well as the important guiding significance to our researches. We have studied comments carefully and have made correction which we hope meet with approval.

Reviewer 1

The manuscript "Relationship between Surface Deformation Displacement and Energy Evolution of Red Sandstone under Uniaxial Compression" provides an accurate analysis of the correlation between surface deformation and energy in red sandstone under loading, revealing the surface deformation process of red sandstone. Based on deformation magnitude and degree, the study identifies four distinct surface deformation stages that correspond to axial deformation stages but exhibit temporal discrepancies, offering important theoretical and technical support for deformation control in free-surface engineering projects such as slopes. However, there remain some minor issues in the manuscript that require revision. Overall, I recommend minor revisions.

Response: Thank you very much for your review of this article and your valuable comments. The author has modified the full text one by one according to your Suggestions. The specific modified content and the reply to the question are shown below.

General opinion

1.Regarding the description of previous unstudied aspects in lines 69-72 within Chapter 2, should this be moved to the Introduction?

Response 1: Thank you very much for your review of this article and your valuable comments. The content have moved to the Introduction.

2. There are numerous formatting issues in the text. Please carefully review and revise them, for example:

Both the figures in Line 383 and Line 388 are labeled as "Fig. 13.";

There is no blank line after Line 383 and Line 388, with text directly following. In contrast, other figures and paragraphs have blank lines in between;

Some parts use "Fig. xx," while others use the full term "Figure," and some are written as "Fig.xx" (without a space);

The format of the title in Line 124, "Fig.1 Test Flow Chart," should be consistent with other figures;

Citation formats in the article should be standardized and unified;

Response 2: Thank you very much for your review of this article and your valuable comments. The line 383 Fig 13 have revised as Fig 14; the line 383 and line 388 have add blank line; all part have changed to “Fig.xx”; in Line 124, the "Fig.1 Test Flow Chart" have changed to "Fig.1 Test flow chart", consistent with other figures; Citation formats in the article have standardized and unified.

3.In Section 3.1.3, where is the analysis of surface displacement changes based on the charts?

Response 3: Thank you very much for your review of this article and your valuable comments. The description and analysis have added in Section 3.1.3.

4.In Fig. 11, some subplots have labeled x-axes, while others are missing labels.

Response 4: Thank you very much for your review of this article and your valuable comments. The image has been modified and the X-axis title has been added

5.Conclusion (2) is overly complex. Please refine it by extracting the key points.

Response 5: Thank you very much for your review of this article and your valuable comments. The conclusions have been reorganized by extracting the key points of deformation in each stage.

Reviewer 2

This paper entitled " Relationship between Surface Deformation Displacement and Energy Evolution of Red Sandstone under Uniaxial Compression" presents investigates the relationship between surface deformation and energy evolution in red sandstone under uniaxial compression using advanced 3D visualization techniques. It is an interesting topic that may inspire the research in rock mechanics. While the topic is relevant to geotechnical engineering, several significant issues need to be addressed.

Response: Thank you very much for your review of this article and your valuable comments. The author has modified the full text one by one according to your Suggestions. The specific modified content and the reply to the question are shown below.

1. In line 21, the "red rock sandstone" should be "red sandstone" (redundant terminology).

Response 1: Thank you very much for your review of this article and your valuable comments. Have modified.

2. In line 27-28, the sentence “According to the amount and degree of deformation” is not clear, please clarify.

Response 2: Thank you very much for your review of this article and your valuable comments. The sentence have changed to “surface deformation of the sample and the degree of change”, which means that the deformation stages are divided according to the amount of deformation displacement on the rock surface and the change rate ( growth rate ) of deformation.

3.In the Introduction section, literature review appears to be somewhat limited in scope. Including more recent and diverse studies could provide a more comprehensive background. Please consider the following papers: 10.1007/s00603-022-02861-4, and 10.1007/s00603-024-03845-2.

Response 3: Thank you very much for your review of this article and your valuable comments. These two papers have been cited with serial numbers of 2 and 26 respectively.

4. Please improve the state in the manuscript such as “Fig.1 Test Flow Chart” and “Research Content and Methods”.

Response 4: Thank you very much for your review of this article and your valuable comments. This part has been enhanced.

5. In lines 174-189, sample preparation description lacks important details about specimen uniformity and quality control

Response 5: Thank you very much for your review of this article and your valuable comments. A description of sample uniformity and quality control has been added, in Table 1 and Table 2, in lines 180 - 182.

6. In Section 2.3, the correlation between energy and surface deformation demonstrated in Equation 8 lacks theoretical foundation, please give more details explanation.

Response 6: Thank you very much for your review of this article and your valuable comments. Relevant theoretical descriptions have been added in lines 255-260.

7. In lines 268-271, failure mode classification lacks quantitative criteria and it seems that the failure modes can be distinguished from Fig. 6.u

Response 7: Thank you very much for your review of this article and your valuable comments. In lines 283-287 , a quantitative description of the damage by the number of cracks is added.

8. The manuscript contains several grammatical errors and awkward sentence structures that need to be addressed. A thorough proofreading is recommended to improve readability.

Response 7: Thank you very much for your review of this article and your valuable comments. The whole article has been proofread.

Reviewer 3/

This paper focuses on the energy evolution process of surface deformation of red sandstone under uniaxial compression, and uses a 3D visualization testing system to observe surface deformation and conduct correlation analysis with the energy changes during the failure process of red sandstone. The manuscript is well written and the topic is interesting and novel. However, some problems were also found in the manuscript. Therefore, minor revisions are required before the paper is published in the journal. All details are summarized below.

Response: Thank you very much for your review of this article and your valuable comments. The author has modified the full text one by one according to your Suggestions. The specific modified content and the reply to the question are shown below.

Required changes:

Point 1: The sentence, “The energy input resulting from construction disturbance (such as tunneling and pile foundation construction) inevitably induces deformation and failure, potentially compromising project safety”, mentioned in the Introduction, are reference for these statement and/or previous works? Please add relevant references.

Response 1: Thank you very much for your review of this article and your valuable comments. Relevant literature has been added to prove that.

Point 2: The descriptions of sample composition in Tables 1 and 2 are irrelevant to the research content of this paper, which focuses on external energy input and surface deformation. It is recommended to remove them.

Response 2: Thank you very much for your review of this article and your valuable comments. The elements and oxides of the samples in Table 1 and Table 2 are the contents of the quality control of the samples. The relevant control descriptions have been added in lines 180-182 to supplement the table.

Point 3: In the Fig. 6 Failure Modes, merely outlines the failure phenomena without providing concrete explanations. It is recommended to provide a more in-depth explanation of the failure modes observed in different samples, along with a thorough analysis of the underlying mechanisms responsible for these phenomena.

Response 3: Thank you very much for your review of this article and your valuable comments. A quantitative description of the damage has been added, and the failure mechanism is described in the discussion section.

Point 4: The axis labels in several figures appear pixelated and difficult to read. The authors should provide higher-resolution versions with clearly legible text.

Response 4: Thank you very much for your review of this article and your valuable comments. Relevant pictures have been modified.

Point 5: The paper contains formatting and content inaccuracies that require correction. It is recommended to further improve them. It is necessary to correct the whole paper for accuracy and precision. Minor grammatical problems in the full text need to be corrected.

Response 4: Thank you very much for your review of this article and your valuable comments. The whole article has been proofread.

---

## [Decision Letter · Decision Letter 1]

10 Jul 2025

Relationship between Surface Deformation Displacement and Energy Evolution of Red Sandstone under Uniaxial Compression

PONE-D-25-28363R1

Dear Dr. Liu,

We’re pleased to inform you that your manuscript has been judged scientifically suitable for publication and will be formally accepted for publication once it meets all outstanding technical requirements.

Kind regards,

Lisong Zhang

Academic Editor

PLOS ONE

Additional Editor Comments (optional):

Reviewers' comments:

Reviewer's Responses to Questions

**Comments to the Author**

1. If the authors have adequately addressed your comments raised in a previous round of review and you feel that this manuscript is now acceptable for publication, you may indicate that here to bypass the “Comments to the Author” section, enter your conflict of interest statement in the “Confidential to Editor” section, and submit your "Accept" recommendation.

Reviewer #1: All comments have been addressed

Reviewer #2: All comments have been addressed

Reviewer #3: All comments have been addressed

2. Is the manuscript technically sound, and do the data support the conclusions?

Reviewer #1: Yes

Reviewer #2: Yes

Reviewer #3: Yes

3. Has the statistical analysis been performed appropriately and rigorously? 

Reviewer #1: Yes

Reviewer #2: Yes

Reviewer #3: Yes

4. Have the authors made all data underlying the findings in their manuscript fully available?

Reviewer #1: Yes

Reviewer #2: Yes

Reviewer #3: Yes

5. Is the manuscript presented in an intelligible fashion and written in standard English?

Reviewer #1: Yes

Reviewer #2: Yes

Reviewer #3: Yes

6. Review Comments to the Author

Reviewer #1: (No Response)

Reviewer #2: The authors have addressed all the comments properly and the quality has been greatly improved. I have no further comments, and the manuscript can be accepted.

Reviewer #3: The authors have adressed all it is recommended for pubilication. it is recommended for pubilication. it is recommended for pubilication.

7. PLOS authors have the option to publish the peer review history of their article (what does this mean? ). If published, this will include your full peer review and any attached files.

**Do you want your identity to be public for this peer review?** For information about this choice, including consent withdrawal, please see our Privacy Policy .

Reviewer #1: No

Reviewer #2: No

Reviewer #3: No

---

## [Editor Report · Acceptance letter]

PONE-D-25-28363R1

PLOS ONE

Dear Dr. Liu,

I'm pleased to inform you that your manuscript has been deemed suitable for publication in PLOS ONE. Congratulations! Your manuscript is now being handed over to our production team.

Kind regards,

on behalf of

Associate Professor Lisong Zhang

Academic Editor

PLOS ONE